# The Post-COVID-19 Era, Fourth Industrial Revolution, and New Globalization: Restructured Labor Relations and Organizational Adaptation

Theodore Koutroukis [1,2,*] , Dimos Chatzinikolaou [1], Charis Vlados [1,3,4] and Victoria Pistikou [1,2]

[1] Department of Economics, Democritus University of Thrace, 69100 Komotini, Greece
[2] Department of Economics and Business, School of Economics, Business, and Computer Sciences, Neapolis University Pafos, Paphos 8042, Cyprus
[3] School of Business, University of Nicosia, Nicosia 2417, Cyprus
[4] School of Social Sciences, Business & Organisation Administration, Hellenic Open University, Par. Aristotelous 18, 26335 Patras, Greece
[*] Correspondence: tkoutro@econ.duth.gr

**Abstract:** This paper explores the directions of adaptation for socioeconomic organizations in the current global crisis and restructuring. We carry out an integrative and critical review, presenting the main questions—and possible directions of response—concerning how the post-COVID-19 era, the fourth industrial revolution, and new globalization seem to affect contemporary labor relations. We focus on the different levels of their manifestation (macro, meso, and micro levels), emphasizing worsening inequality trends in the work environment and the resulting organizational readaptation that seems to be required nowadays. The restructured labor markets can benefit from the diffusion of institutional innovations based on integrated social partnership schemes at the macro–meso–micro levels. We emphasize organizational adaptation at the microlevel, as the innovation and change management mechanisms it enables, presupposes, and harnesses are imperative for exiting any crisis.

**Keywords:** post-COVID-19 era; fourth industrial revolution; new globalization; restructured labor relations; institutional innovation; social partnership; organizational adaptation; change management; innovation; green transition

## 1. Introduction

A new structure for the global economy seems to crystallize in the emerging post-COVID-19 era progressively [1,2]. Simultaneously, saturated professions and industries appear to be profoundly restructured by the fourth industrial revolution (4IR), which seems to have been greatly accelerated by the pandemic [1]. During COVID-19, severe problems in international supply chains in many sectors of global production emerged, creating the substrate for stagflationary trends. Amid these global restructurings, Russia's illegal invasion of Ukraine in 2022 further escalated inflation and the required transition to new energy corridors; simultaneously, it seems to have triggered shifts in the Western "modus operandi" [2,3].

Therefore, organizational survival and growth in the face of such a significant paradigm shift for a wide variety of socioeconomic entities worldwide (private, public, and mixed) is perhaps the most crucial contemporary challenge [4,5]. Rapid digital transformation and continuous organizational adaptation for sustainable growth seem to be the main challenges for the near future of all socioeconomic organizations nowadays [6–8]. Thus, less competitive firms—in broad terms regarding internal innovation compositions—seem to face particular deficiencies in their adaptive capacities [9,10]. For example, the COVID-19 crisis seemingly exacerbated specific organizational weaknesses due to a lack of progress in sustainable development goals [11]. However, it remains relatively unclear how we could

formulate guidelines for adaptation in the post-COVID-19 era that apply to all socioeconomic organizations. This paper attempts to fill this gap in the literature by conducting an integrative and critical review of published research focusing on organizational adaptation for the post-COVID-19 era [12,13]. It also consolidates, captures, and analyzes those related dimensions concerning the current transformation in work relationships from a cross-spatial (macro, meso, and micro) perspective [14]. This critical review aims to elliptically present the main questions concerning the post-COVID-19 organizational transformation by highlighting possible gaps in our understanding and suggesting directions for business adaptation to the new conditions [7,15].

The remainder is structured as follows. The second part examines the literature that explores aspects of the ongoing global crisis. It arrives at the new globalization approach, i.e., another perspective of the global evolutionary transformation besides the post-COVID-19 era and the 4IR. A subsequent part integrates these global restructuring horizons into the labor market (mesolevel) by discussing emerging disparities in contemporary work relations and all socioeconomic organizations (microlevel). It discusses the constituents of a new social dialogue to reinforce macro–meso institutional reorientations and the facets of the required organizational adaptation. The last part concludes and discusses the research prospects derived from the analysis.

## 2. Literature Review

### 2.1. The Prolonged Global Crisis: Where Are We Nowadays?

The worldwide financial disruption of 2008, the pandemic of 2019–2020 that accelerated the fourth industrial revolution, and the Russian invasion of Ukraine in 2022 that hastened the ongoing energy transition seem to be milestones of the emerging new globalization [16,17]. The old regime seems to have finally given way to a new reality, which appears to be gradually settling on the structural maturation of the previous globalization phase, which lasted from 1980 to about 2008 [16]. Figure 1 captures the economic background of this transmutation period—a gradual transition to the new globalization [18,19]—by emphasizing recent macroeconomic data.

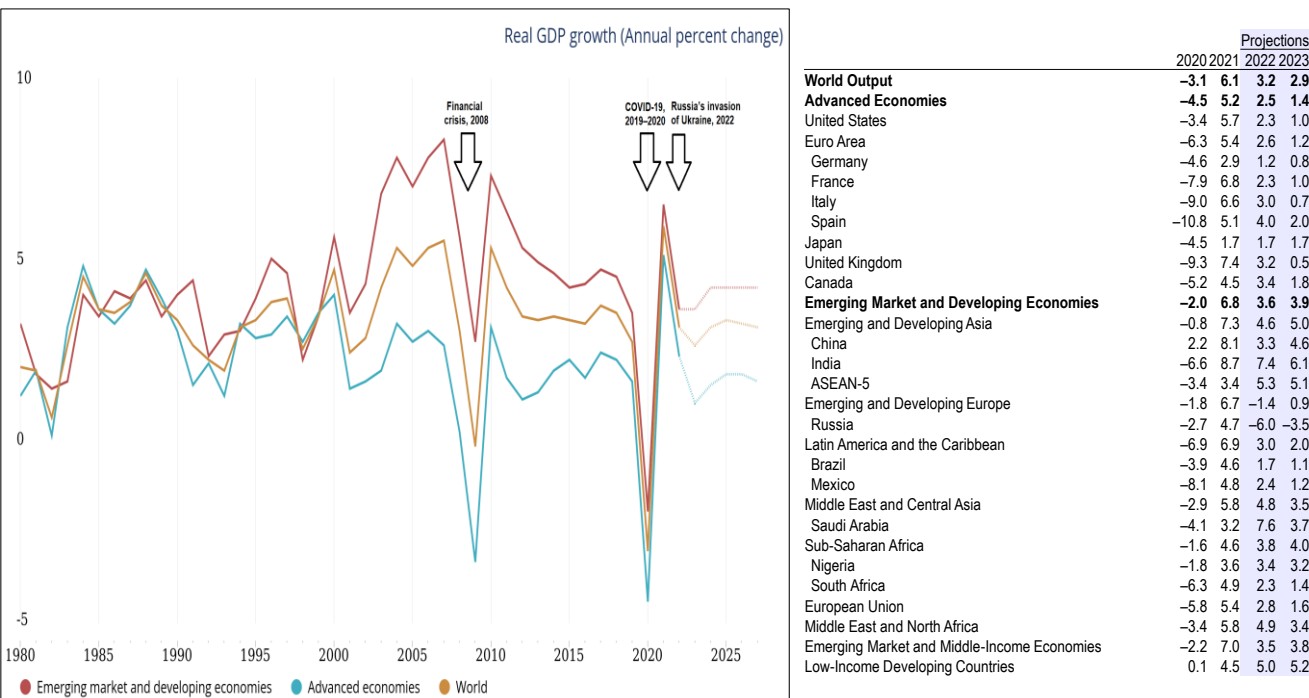

**Figure 1.** Capturing the global crisis in macrofigures. Source: Based on [20].

Considering the three major global crises in the 2000s, we observe that real GDP growth has been moderate from 1980 up to this date. The financial crisis of 2008 and the pandemic of 2019–2020 have led to two significant reversals that have affected the global economy. Emerging economies were less influenced by the 2008 crisis (compared to 2020), a trend that appears to have exacerbated earlier geopolitical tensions. The economic setback of the emerging economies (primarily China) seems to have revitalized tendencies that remained previously silent due to robust growth rates and the promotion of global trade [21].

The pandemic crisis of COVID-19 appears to have been the most critical challenge the global economy has experienced since World War II (WWII) [22]. The pandemic's impact was significant, shaking investor and consumer confidence as all economies entered a sharp recession after relatively low growth rates in the 2010s [23]. We observe an unprecedented slowdown in 2020, indicative of these adverse developments. In particular, the world output decreased by 3.1% and then increased by 6.1% in 2021.

During the COVID-19 outbreak, approximately 400 million full-time employees were deprived of their positions due to the lockdown measures of 2020 [24]. This unprecedented disruption exerted significant influence on all industries, sparking a debate on whether adverse developments would soon occur, such as the exacerbation of underdevelopment, primarily across the developing economies, a rise of political extremists, or an increase in geopolitical tensions [25–28]. Unfortunately, these fears appear to have been confirmed nowadays, culminating in Russia's invasion of Ukraine in 2022, which ignited a recession, the magnitude of which most developed economies had never faced after WWII. For example, the inflation rate for the Euro area was 7.3% in 2022 and is expected to be approximately 3.9% in 2023 [20]. Russia's economy appears significantly affected after the invasion that led the Western allies to impose severe sanctions and economically isolate this country as a perpetrator of actions in flagrant violation of international law [29].

Therefore, even though a V-shaped recovery seemed possible amid the pandemic crisis, today's phenomena point in a different direction [30–32]. Some scholars warned that the most likely immediate scenario for the global economy was L-type growth, especially for the less developed and weak economies [19,33,34]. This projection was made because less-developed socioeconomic systems are usually not as flexible and cannot benefit from the recovery of international trade as much as their developed counterparts. Russia's invasion that triggered a world war with unpredictable conclusions reaffirms an L-type development trajectory for the global system. Most conventional growth models that forecasted a direct V-shaped recovery of the GDP for all the affected countries by considering a few ceteris paribus parameters proved once more inadequate to predict where we are headed [30–32]. As Figure 1 shows, based on the scenarios put forward by the IMF [20], no robust growth is expected soon. Therefore, we deem it critical to examine the development dimensions that could illuminate specific evolutionary trends and help predict the global socioeconomic system's new trajectory. As the next section discusses, the world appears to be heading toward an irreversible rebalancing, accelerating the 4IR, which affects all levels of our socioeconomic symbiosis.

### 2.2. The Accelerated 4IR and the New Globalization

In the terminology suggested within the 4IR theoretical horizon, today's global socioeconomic system faces drastic technological transformations that will soon lead to an entirely new reality [35]. In attempting to study the phenomena of global mutation, the 4IR approach builds on evolutionary contributions such as Schumpeter's [36] or Kondratieff's [37]. This analysis perceives development as discontinuous and transformed over long waves of growth and recession—considering techno-economic paradigms, according to Perez's neo-Schumpeterian contribution [38].

However, the relevant scholarly debates and policy forums started to use the 4IR only recently. We find its first mention as "Industry 4.0" in a paper published by the German government in 2011, aiming to describe a high-tech strategy initiative [39]. In

2015, Schwab [40] helped to diffuse the term "Fourth Industrial Revolution" in a piece written in *Foreign Affairs* that discussed the outcome of the Davos World Economic Forum Annual Meeting. In 2016, Schwab [41] authored a homonymous nodal textbook, laying down the respective definitions of the 4IR. In particular, Schwab [41] contends that the first industrial revolution refers to the mechanization of manufacturing via the use of steam engine applications, the second to mass production through novel electricity usages, and the third to information technology based on extensive automated processes. According to Schwab [41], the primary challenge to deal with is the ongoing digital revolution, as there is an increasing fusion between the natural–biological and digital worlds—the most surprising phenomena acknowledged by the 4IR scholars are the "cyber-physical systems" which appear to fuse elements from the tangible and intangible world [42,43]. In Schwab's [41] view, humanity's response to this new challenge must be all-encompassing and multidimensional, involving private actors, universities, and civil society. Schwab [41] also stresses that the 4IR may increase incomes worldwide and diffuse cost-efficient goods and services; the demand side also appears to be strengthened as this global socioeconomic metamorphosis increases consumer options. However, Schwab [41] also urges not to ignore the possibility that these new advances will likely cause an escalation in societal tensions and the replacement of manual labor by automated systems. Therefore, as Schwab [41] contends, the changes that innovation causes cannot be easily predicted, and, as a result, new firms will disrupt incumbents more quickly than previously.

The new globalization is another comprehensive perspective that describes the global system's evolutionary transformation, investigating the prerequisites behind today's prolonged global crisis. According to Chatzinikolaou and Vlados [16], we can interpret the world's socioeconomic evolution after WWII by building on the synthesis of three relatively distinct theoretical platforms. Table 1 presents how these three lead to acknowledging different post-WWII phases [44,45].

**Table 1.** The emerging new globalization is the recent phase in the global system's evolutionary trajectory.

| | **Hegemonic Stability Theory: International Regime** | **Regulation School: Development–Crisis Theoretical Platform** | **Evolutionary Socioeconomic Approach: Generations of Innovation** |
|---|---|---|---|
| **1945–1973: First postwar international growth and national development period** | US hegemony and the power of bipolarity | Fordist growth | Aggregative innovation |
| **1973–1980: Crisis and pre-globalization period** | Bipolar system's crisis | Fordist crisis | Combinational innovation |
| **1980–2008: Globalization period** | Gradual transition to the post-Cold war period | Globalized post-Fordisms | Integrated innovation |
| **2008–New or restructured globalization period** | Seeking a new multipolarity | Searching for realistic hybrid post-Fordisms | Quest for an organic, ecosystemic, and open innovation |

For the hegemonic stability theory, the international system leans toward relative balance when one superpower prevails over the rest [46–50]. Second, the "École de la Régulation" is a stream developed in the early 1970s to explain the structural behavior of socioeconomic systems—primarily from a nation-centered and neo-Marxist perspective [51–54]. Also, the evolutionary socioeconomic approach investigates the economy that never rests due to the endogenous forces of innovation that structurally transform all socioeconomic systems and actors. Certain scholars also criticize and reject the standard and conventional neoclassical maximization rationale, arguing that the firm is not a "black box" but a "living organism" [55–59]. Based on these theoretical foundations, Rothwell's [60]

work for the different generations of innovation matches the global evolutionary trajectories with innovational developments within the organizational boundaries.

Therefore, the new globalization approach appears to delve into the underlying structures apart from the technological changes that the 4IR perceives behind the world's transformative evolutionary trajectories. The new globalization is a relatively recent concept. Most scholars express a rather "pessimistic" view of tomorrow's global system, as it appears that the new globalization will bring—and perhaps has already brought—increasing worldwide tensions. For example, we see increased stress between industrialized and African nations [61] or a return to Keynesian-type protectionist policies and the rise of BRICS (Brazil–Russia–India–China–South Africa) as alternative safety hubs [62]. We also see further polarization amid today's global hiatus [63], a reinforced form of adverse multipolarity, frequent changes in regulation and politics [64], or the prevalence of worldwide corporations against governments [65].

Chatzinikolaou and Vlados [16] suggest examining scenarios for the new globalization that are not necessarily adverse—see also Baldwin's [66] and Roach's [67] approaches. Chatzinikolaou and Vlados [16] think that we are amid a transitional period, the outcome of which has not yet been crystallized. Today's phase started approximately in 2008—the financial crisis' aftermath. It now leads to seeking a new multipolarity in international relations, searching for realistic hybrid post-Fordisms [1] concerning a platform that favors sustainable development and the quest for organic forms of innovation within firms. According to Chatzinikolaou and Vlados [16], how the global community responds to these quests will determine whether this emerging regime will crystallize in better socioeconomic performance—a new realistic global liberalism—or lead to further nation-centric fragmentation—the worst scenario. Chatzinikolaou and Vlados [16] suggest that we are currently in a medium-level state of affairs where restructured forms of multipolarity seem to prevail worldwide.

How can organizations innovatively adapt to this new global socioeconomic environment? The COVID-19 crisis seems to have reignited earlier geopolitical tensions, rightly magnifying the Ukrainians' struggle against Russia, whose invasion in 2022 was a culmination of the pressures that emerged in the post-COVID-19 era. Scholars of this stream had already warned—from COVID-19's outbreak—that this pandemic crisis could reawaken older conflicts due to its immediate impact on all economies and the requisite extremely protectionist measures to avoid the pandemic's effects [18,68]. The global system's most profound challenge today is promoting innovation, which is requisite for exiting all crises. We also think that digital transformation, the optimal strategy for business growth according to most 4IR scholars, seems largely unattainable for less developed organizational systems [69,70]. The study by Modiba and Kekwaletswe [71] is an example that appears to confirm this claim, as they argue that there is limited technological accessibility in Sub-Saharan Africa and other countries with a rural population. Although Modiba and Kekwaletswe [71] observe a trend toward digitization (43 percent of adults in Sub-Saharan Africa already have a bank account compared to 34 percent in 2014), they believe that technological, organizational, and environmental issues come before digital transformation.

Therefore, we must cautiously investigate today's imperative of organizational readaptation, as not all actors have equal development opportunities. In Maynard's [72] view, navigating this new phase necessitates significant public-private partnerships and, at the same time, new effective multilevel (local, national, regional, and global) change management mechanisms are needed [10]. Otherwise, the wealth gap will widen, geopolitical conflicts will escalate, and sustainable development initiatives will be undermined; if we do not respond to these challenges, our world will face a repetition of the disharmonies and pitfalls that marked the decline of past industrial "evolutions". From a critical perspective, the following section examines the facets of today's restructured working environment, attempting to suggest organizational adaptation guidelines for all socioeconomic organizations.

## 3. Discussion: Restructured Labor Relations, Institutional Innovation, and the Imperative of Organizational Adaptation

*3.1. Post-COVID-19 Work Environment, Emerging Labor Inequalities, and Macro–Meso Adaptations*

As the global socioeconomic system undergoes significant changes, new problems and obstacles seem to emerge in labor relations—i.e., mesoenvironment implications. The COVID-19 crisis appears to have further hastened the labor market's restructuring, mainly due to a wide dispersion of remote work across various industries and organizational environments [73]. Also, this crisis did not affect all social categories and strata to the same degree. According to Pattenden et al. [74], mainly the vulnerable were hit and remain insecure in the post-COVID-19 era. Today's work environment seems to have resulted in a widening income gap between skilled and unskilled occupations—we also observe discrepancies among formal and informal jobs or small-business manufacturing and large corporations that exploit scale economies [75,76]. We also see worsening trends in the post-COVID-19 era for female workers in retail trade, accommodation, food services, and other low-value-added occupations [77].

This era of digitalization appears to lead toward a clear division within the working class: between the poor and wealthier employees, making the problems of social inclusion and labor market disparities sharper [78]. In this direction, an essential point for today's restructured work environment is how national labor regulations and institutions mitigate these issues. Most relevant laws and policies seem to have been built around the traditional workplace that facilitates the action and organization of workers in trade unions and other work councils. Thus, the novel development of multiple and decentralized workplaces could undermine trade union rights, activity, and power, transforming the labor-management balance in favor of the employer [79]. In the post-COVID-19 era, governments appear to be willing to pass labor support measures by continuously boosting the demand side [80]; however, we think that supporting supply with skill development and training—and hence innovation—is critical as the digital divide seems to widen [81].

In particular, the COVID-19 crisis significantly sharpened income and growth inequalities [82]. The informal labor market (black or grey) was among the gravest hit sectors, as we observed an overall drop in labor earnings [83]. Furthermore, a significant part of the workforce was obliged to pay higher health and care costs amid the pandemic (a trend that appears to worsen nowadays) as a means of protection or recovery from the virus [84]. All these adverse developments are evident from notable trends in social and labor market inequalities—primarily in Western countries but also in emerging economies, with a more significant effect. We have observed an increase in unemployment, a fall in the rate of participation in the labor force, and a rise in the "working poor", meaning those population segments whose incomes have fallen below the extreme poverty threshold [77,82,85].

Therefore, labor market disparities will also emerge more robustly in the upcoming years. In a micro-oriented analysis, a widened gap of sophistication and intellectual capital among employees is a direct consequence of the acceleration of the 4IR nowadays (e.g., the diffusion of "cyber-physical" systems) [86]. Also, rising health inequalities are another facet of the challenges we must face in the post-COVID-19 labor market [87,88]. The dialogue on health and safety in the work environment appears to have grown significantly due to the pandemic, with a considerable part of the more capable workforce now looking for remote jobs. In contrast, less sophisticated workers face increasingly various sanitary risks in their traditional outside-of-home occupations [89].

At the sectoral/industry level, the relevant landscape has been much different. The majority of businesses have quickly adapted to the new circumstances. Employers' decisions have impacted many collective bargaining agreements regarding working conditions, compensation, benefits, and personnel reductions. The nature of the negotiation process allows for the adaptation of employment relations considering the concerns of the labor side. That kind of bargaining at the industry level may help employers establish a common goal with employee representatives to address the current challenges affecting the labor

market [90]. The potential of worker participation in post-COVID-19 social and economic life has been related to productivity, innovation, and efficiency [91].

On the other hand, the enterprise level is the most appropriate to promote flexible partnership schemes to add value to human resources functions and improve employee relations. Thus, social partners or management should undertake similar initiatives to promote joint consultation schemes at the workplace. That decentralized microlevel would be instrumental in enacting an employee voice by participating in consultation procedures within any firm. Notably, it will be fruitful to enhance the labor-management concertation at the business level [79]. The successful function of such partnership schemes worldwide could be guaranteed, inter alia, by implementing long-lasting social dialogue institutions, fostering pertinent tailor-made training programs for the stakeholders' representatives, and demonstrating good partnership practices to reduce industrial conflict and add value within enterprises [92].

Bipartite or tripartite concertation among the parties involved (employees, employers, or the government) can foster relevant and effective policy measures by preventing worsening social and labor market inequalities [93]. Nevertheless, the success of these interventions appears moderate to date. For example, most EU countries did not seem to have used social partnership procedures in planning and implementing measures against COVID-19 [94]. Previous experience has proven that partnership and employee voice schemes can effectively confront industrial conflict and labor market inequalities, especially at the microlevel of firms. Therefore, despite recent governmental efforts—primarily in Western countries—to reinforce consumer spending and offer respective labor market provisions, state intervention must seemingly build on new social dialogue schemes and concertation [94]. Such synthetic and integrated partnership strategies seem necessary to promote or implement effective economic and social policies. Certain studies have underlined that when policymakers wish to succeed in their plans, they have to convince the parties concerned—for example, employees, employers, trade unions, and governments—that their interests and opinions have been considered during the pertinent decision-making process [95]. The Copenhagen Centre's [96] nodal definition discusses a similar perspective, arguing that social partnership refers to "people and organizations from some combination of public, business, and civil constituencies who engage in voluntary, mutually beneficial innovative relationships to address common social aims through combining their resources and competencies". Therefore, bipartite, tripartite, or quadripartite concertation among the parties involved may foster effectiveness in policy decisions and help avoid worsening social and labor market inequalities. For example, many European countries took employment protection measures amid the pandemic crisis with the social partners' direct or indirect involvement [94]. However, as Figure 2 illustrates, these policy approaches incorporate the promotion of institutional innovation only to a minor extent.

Institutional innovation refers to how a socioeconomic system's cognitive, regulatory, and normative bases are changing [97]. When institutional innovations acquire utility and legitimacy, they become "institutionalized" and overcome frictions or resistance altogether [97]. In the current global adjustment, promoting more effective forms of social dialogue is imperative for a restructuring system to be institutionally adaptive [98,99]. However, this understanding primarily needs a vision toward structural reforms, leading to the building of inclusive institutions that secure private property rights, encourage investment in new technologies or skills, and distribute power pluralistically. Otherwise, institutions are driven toward becoming extractive, with elites extracting resources from the lower social strata and development incentives being relatively absent, along with state control being significantly concentrated in the elites [100]. This constant tug-of-war occurs mainly in less advanced macro–meso–micro socioeconomic systems, which must proactively adapt to the current dynamics of the new globalization like all the rest. However, that realistic, innovative liberalism can only be achieved globally by the effective public–private partnerships (primarily meso–micro) that exhibit the following characteristics in all national

socioeconomic systems to some extent. Public interventions must boost innovation, support local development, and provide free access to education.

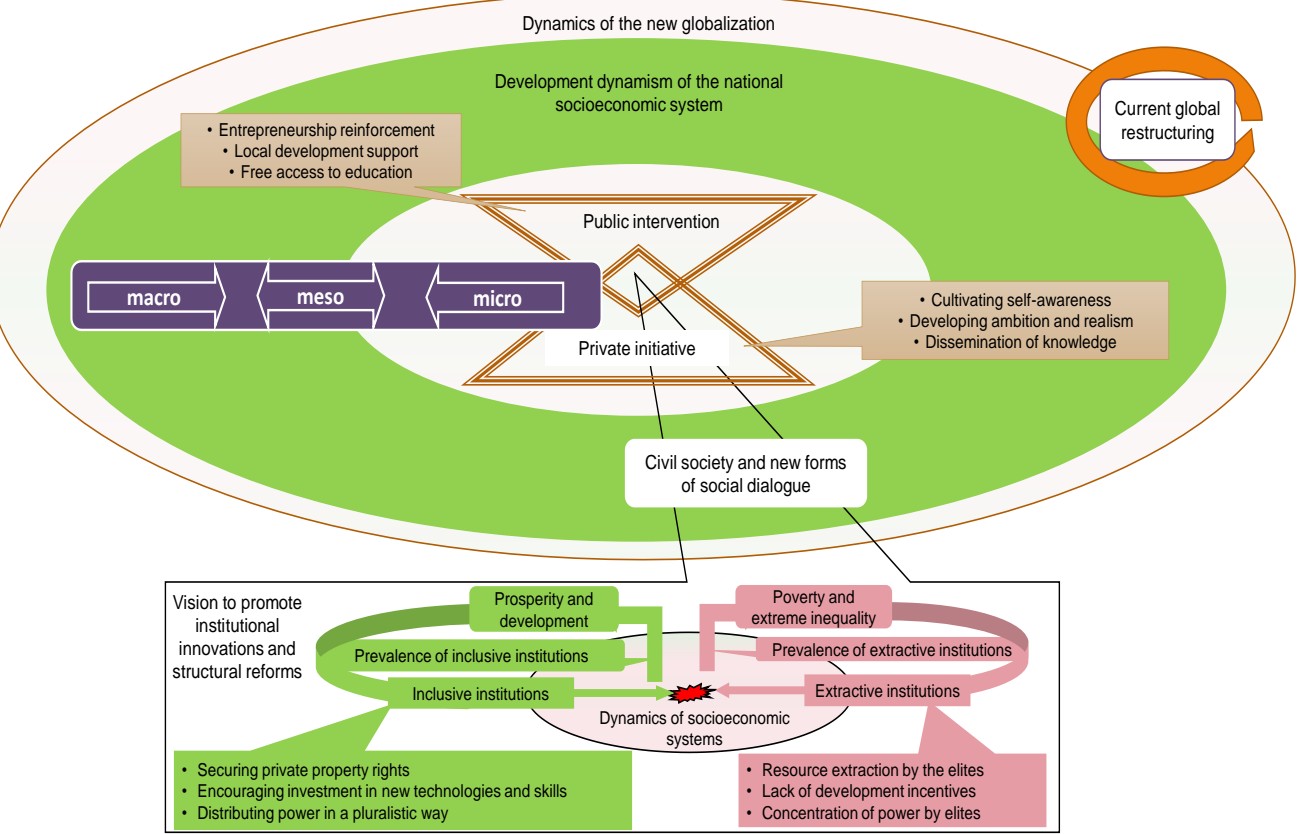

**Figure 2.** Shaping an institutionally adaptive socioeconomic system through new forms of social dialogue.

Therefore, building new multipolar mechanisms capable of producing institutional innovations to combat pressing issues is critical for the emerging international regime [101]. Organic innovation is the main criterion for determining whether there will be a sufficiently stabilizing and balanced system in the new globalization. Regardless of the spatial level, all socioeconomic organizations must systematically invest in their evolutionary strengths, while also healing their weaknesses [16]. We are experiencing an evolutionary and ongoing transition concerning global capitalism, labeled as the 4IR, post-COVID-19 era, or new globalization. As historical reality proves, entrepreneurial and institutional innovation is the force that continuously overturns the various equilibria and acts as the real revolutionary force in the history of all capitalist transitions [19].

Therefore, in this transforming environment, creating more sustainable labor markets requires resilient institutions that provide social protection that encourages and supports structural adjustment [102]. Present-day policies should be reliable and inclusive, with citizens having access to them when they are needed. These policies should aid all, especially those without other sources of support. In addition, policies should help workers adapt to change in a way that is sustainable in the long term. Given the current funding constraints, it is vital that social protection policies not simply finance consumption but have a substantial investment component. For example, instead of compensating individuals according to their past efforts, policies should help them reintegrate into new activities [102].

Thus, private initiatives must constantly cultivate their self-awareness, develop realistic ambitions, and disseminate new knowledge [103]. In the next section, we look at specific

micro-level aspects—e.g., human resource management (HRM)—that seems requisite for adapting to this new working and institutional environment.

*3.2. Toward a Post-COVID-19 Organizational Readaptation*

Concerning microlevel effects, we discern profound transformations in the post-COVID-19 era. For example, less personal contact and professional relationships with colleagues seem normal in the newly emerging working environment [104]. Also, the fact that remote work is now part of the daily routine for many service sector employees seems to lead toward less innovation, as home workers do not appear to be seeking increased contact with their colleagues, which could drive improved performance [105]. This digital transformation challenges the foundations of employee commitment toward the organization's vision, primarily due to this relative lack of effective interpersonal communication [106]. Research has shown that collective bargaining agreements—considering labor aspects, such as working conditions, compensation, benefits, or personnel reductions—are restructured nowadays to deal with this internal transformation, targeting shared employee–employer goals to reinforce organizational productivity and efficiency [90,91,107]. However, financial uncertainty seemingly has pushed many companies to mutate their employer–employee relationships into business-to-business agreements to cut down tax expenditures, regulations, and personnel benefits [108,109]. [2]

Primarily, employers must take organizational-adaptation action that supports employee well-being by paying attention to health and safety, [3] monitoring employee performance, modifying working schedules if necessary, and supervising employees regardless of their work location [110]. Employers also can choose relocation to decentralized low-cost workplaces or abroad to cut off production costs. [4] Alternatively, they can promote flexible partnerships—new joint consultation schemes at the workplace—that improve employee relations and add value to specific human resource functions [111,112]. In this transforming global and organizational landscape, HRM practitioners must understand their critical role in helping enterprises maneuver this crisis. They must be oriented toward boosting their workers' morale and engagement while working from home, developing their soft skills, and providing assistance in managing stress [113]. If they fail to follow these duties and priorities, unwanted results will probably appear, such as limited job satisfaction and imbalances between cost reduction and employee productivity [114]. Other optimal post-COVID-19 HRM practices involve digitizing the recruitment processes, driving employees and executives to foster the necessity of change, emphasizing upskilling, and building a capacity for crisis management skills [115,116].

The literature on reorganized labor relations and the 4IR do not appear to suggest a general framework of organizational adaption that could concern most socioeconomic organizations by focusing on the critical constituents of innovation to exit the crisis. As a biologically related concept, [5] organizational adaptation is about external environmental selection (stimulus) and internal variations (choice), as some of the founding scholars of this stream suggest [117]. In the post-COVID-19 era, we observe a significant environmental transformation that requires reintegrated organizational procedures, primarily oriented toward digital evolution [118,119]. According to the developments that have taken place so far, this research stream to the post-COVID-19 organizational adaptation appears to be in its early phases. It requires a fusion of generic guidelines that could illuminate facets of the internal business environment that lead to innovation—the only exit from any crisis [10]. Thus, we suggest an organizational refocusing to address today's pressing problems emerging in the new globalization based on fundamental principles in the practice and theory of change management; Figure 3 illustrates the suggested conceptual framework.

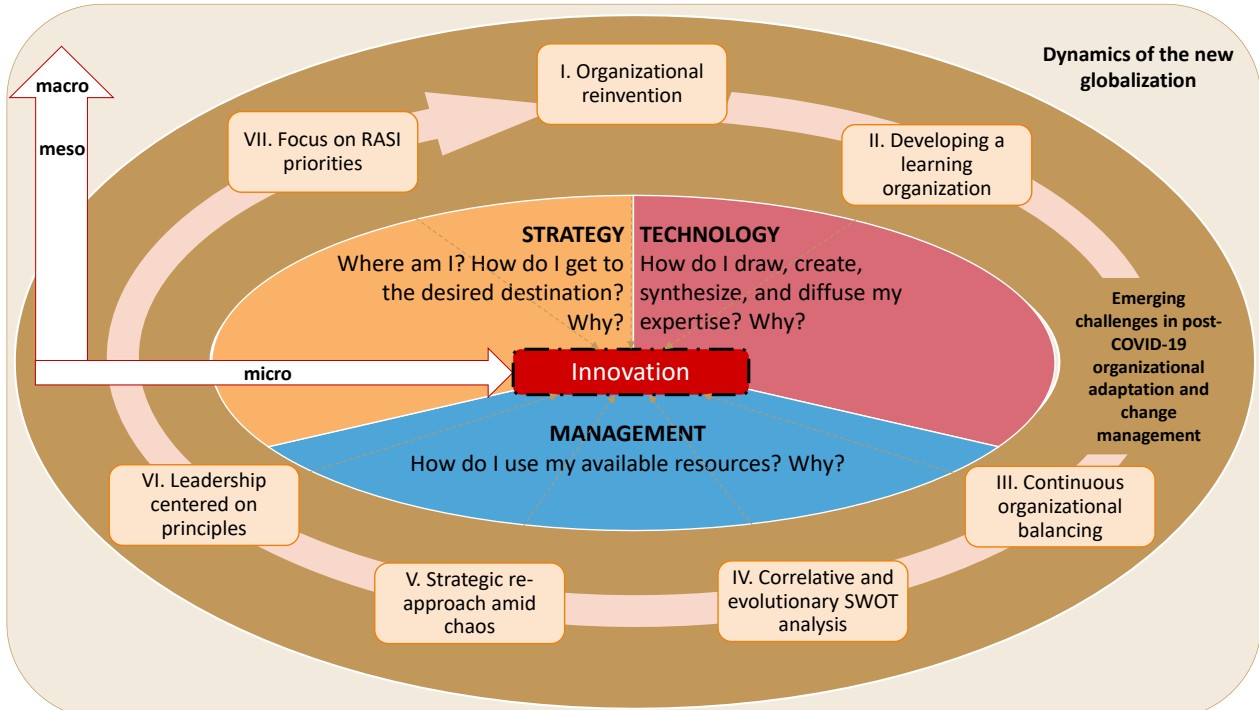

**Figure 3.** The suggested integrative approach to organizational readaptation.

We suggest innovation is the heart of an organization's evolving core, generated by effective strategy–technology–management syntheses (the Stra.Tech.Man approach, see [59,120,121]). The successful Stra.Tech.Man synthesis depends on how the socioeconomic organization meets the prerequisites toward a series of profound and critical questions. Strategy is about "Where am I? How do I get to the desired destination? Why?" Technology results from "How do I draw, create, synthesize, and diffuse my expertise? Why?" Management corresponds to "How do I use my available resources? Why?" The literature also suggests investigating the change-management mechanisms that functioned as preconditions for these innovational forces to emerge [122]. Therefore, we think the following fundamental theoretical concepts of microlevel organizational transformation are critical for helping all socioeconomic organizations navigate today's emerging macro–meso restructurations (e.g., transformed labor relations in the post-COVID-19 era and the new globalization).

I. Organizational reinvention: according to Goss et al. [123], reinventing the organization is a journey filled with necessities and difficulties—those who ride the "roller coaster" must be prepared for a challenging process that unveils and changes the hidden decision-making assumptions and foundations. Organizations usually fail to reinvent themselves because they base their future development only on past practical experience. According to Goss et al. [123], doing the same thing repeatedly and expecting different outcomes is meaningless—an issue encountered in complacent managers who overconcentrate on the organization's prior belief system. Against this backdrop, Goss et al. [110] argue that redesigning the organization entails engaging in thorough and methodical selfcriticism rather than responding like a frightened aristocracy that must restore previous "certainties";

II. Developing a learning organization: regardless of hierarchy or experience, all employees are accountable for finding, analyzing, and addressing practical problems, allowing the organization to ameliorate itself continuously through experimentation and relearning. According to Senge [124], humanity's real challenge is our failure to comprehend how we should govern human systems, suggesting that we need "learning organizations" that are sensitive to change and not as bureaucratic

in their internal environment. Senge [124] emphasizes that hierarchy persists even in highly networked enterprises. However, according to Senge [124], the hierarchical distribution of power in learning organizations differs significantly from that of traditional enterprises because this chain of command comprises mostly guiding concepts and ideas;

III. Continuous organizational balancing: according to Duck [125], the management paradigm that is usually appropriate for day-to-day procedures is insufficient for handling change since it is analogous to a patient receiving five surgical operations simultaneously, which might result in death due to shock. Breaking down the requisite organizational transformation into smaller parts seems insufficient, as the administration must deal with the dynamics of change; the real problem is bringing innovations into the intellectual work arena primarily while simultaneously balancing all the organizational jigsaw pieces. Employees frequently do not believe in the excellent outcome of change—they tend to be skeptical of the organization's new route, trusting organizational change only when they observe an action and related results that demonstrate the benefits of a change program. Overall, Duck [112] contends that this "art of balancing" implies that change management is a collective endeavor that all stakeholders must do;

IV. Correlative and evolutionary SWOT analysis: according to Vlados [120], all organizations must exploit their comparative strengths to innovate, capitalizing on opportunities that arise through time; accordingly, they must avoid the possible threats that emerge from their "correlative" and evolutionary weaknesses. There are no equivalent opportunities and threats for all socioeconomic organizations; these must be continuously reexamined in light of the specific strengths and weaknesses developed over time. From Vlados' [120] viewpoint, the correlative SWOT means finding historically constructed comparative strengths and weaknesses that lead to potential opportunities and threats based on understanding the organization's strategy, technology, and management foundations;

V. Strategic reapproaching amid chaos: according to Kotler and Caslione [126], the world has entered an irreversible disruption, and organizations need to reapproach the mechanisms by which they assess their performance. According to these authors [126], many organizations confuse goals with means and processes with outcomes, resulting in significant inefficiencies when trying to cope with the ever-unfolding crises in the emerging era of chaos. Kotler and Caslione [113] propose establishing an integrated cycle of implementation and a framework for executing strategies that cope with "chaos" by stressing the organization's continual restrategizing to gain an edge over rivals;

VI. Leadership centered on principles: according to Covey [127], as multiple defensive mechanisms resist organizational transformation, all stakeholders must be prepared to express how they think the organization acts within its external environment. The goal is to understand what constitutes the consciousness behind organizational actions, and, in this context, the principles and beliefs managers have about the strategy are less significant than the unconscious methods applied on the ground. Overarchingly, Covey [111] argues that principle-centered leaders recognize that leadership entails embracing change that necessitates uncovering and committing to the organization's fundamental beliefs;

VII. Focus on RASI priorities (resilience–adaptability–sustainability–inclusiveness): in another research paper that we are working on during this period, titled "Green organizational reorientations for the new globalization", we suggest that organizations must effectively synthesize a series of internal dimensions to deal with the new global reality. We suggest that innovational greenness can be achieved through the organization's environmental, social, and corporate governance, aiming toward the synthesized goals of resilience, adaptability, sustainability, and inclusiveness. This green targeting seems requisite, especially considering the present-day energy

transition accelerated by Russia's invasion of Ukraine. However, although the relevant literature acknowledges the significance of these organizational goals, little integration is observable [128,129]. Therefore, change management and innovation residing upon these synthesizing principles could help all socioeconomic organizations in the new globalization.

## 4. Conclusions and Prospects

This integrative review aimed to elliptically present the main questions concerning the seemingly required transformation of organizations amid today's global crisis: the post-COVID-19 era, 4IR, and new globalization. We used this global restructuring as a canvas to highlight the apparent gaps in our understanding of organizational transformation by considering current mutations in labor relations and putting forth guidelines that can help most socioeconomic organizations navigate the crisis. This critical inquiry concludes by considering the following recapitulating observations that assist with the argument supported throughout the text.

(A)  The COVID-19 crisis exacerbated preexisting trends, crystallizing a transition toward the 4IR. We think that an L-shaped type of global growth and development will be the definitive worldwide conclusion in the short run, expressed as persistent stagflationary pressures. The hastened energy transition sparked by Russia's invasion of Ukraine–followed by geopolitical reevaluations across all players globally–is another significant milestone that we think belongs to the new form of globalization that is emerging nowadays. Further research seems imperative in coevolving concepts of global evolutionary transformation, such as the post-COVID-19 era, the 4IR, and the new globalization.

(B)  The above phenomena, in conjunction with the outbreak of the COVID-19 pandemic, have worsened labor inequalities. Most EU countries did not adequately use social partnership procedures to plan and implement measures against COVID-19 and adjust them to the new context of the economy at the national, industry, or company level. Nonetheless, previous experience has proven that partnership and employee voice schemes can effectively confront industrial conflict and labor market inequalities, especially at the microlevel of enterprises;

(C)  The accelerated 4IR can also explain why the labor market is profoundly restructured these days. Digital transformation is essential for all socioeconomic organizations nowadays. To some extent, developing "cyber-physical" systems is a necessity—particularly for less adaptable employers and employees—as they can only be built on top of new knowledge and soft skills. However, avoiding new forms of exclusion and labor inequalities caused by digital technologies is undoubtedly a significant emerging challenge. Most countries have not used effective social partnership schemes to deal with today's global transitional period. Thus, additional research on formulating socioeconomic policies to exit the crisis seems to be needed based on frameworks that promote institutional innovations and adequate macro–meso–micro public–private partnerships;

(D)  Additionally, developing new organizational forms of environmental awareness—simultaneously, at the level of central management and employees (from top to bottom and vice versa)—seems to be an essential adaptive feature for organizations aiming at their own innovative greenness. The hastened energy transition (see the war in Ukraine) seems to confirm this central environmental concern and the need to promote organic innovations. The green change management dimensions we have introduced could trigger further studies addressing climate change through mechanisms that promote organizational resilience, adaptability, sustainability, and inclusiveness;

(E)  We finally consider the microlevel potential of organizational adaptation as an inextricable—and perhaps the most significant—link to all organizations' survival and development. This paper supported the argument that innovation is the only way to exit the crisis based on adequate change-management mechanisms. We think further

research could better integrate present-day organizational transformation programs into the fundamental theoretical perspectives of change management. The conceptual scheme we introduced in Figure 3 can be used as a compass for change practitioners in these turbulent times and help socioeconomic organizations improve their strategy, technology, and management.

**Author Contributions:** Conceptualization, T.K., D.C. and C.V.; methodology, T.K., D.C. and C.V.; formal analysis, T.K. and D.C.; investigation, T.K. and D.C.; resources, V.P.; data curation, D.C.; writing—original draft preparation, T.K. and D.C.; writing—review and editing, D.C., C.V. and V.P.; visualization, D.C.; supervision, T.K., D.C. and C.V.; project administration, T.K., D.C. and C.V.; funding acquisition, T.K. and V.P. All authors have read and agreed to the published version of the manuscript.

**Funding:** This research received external funding by the Neapolis University Pafos.

**Institutional Review Board Statement:** Not applicable.

**Informed Consent Statement:** Not applicable.

**Data Availability Statement:** Not applicable.

**Conflicts of Interest:** The authors declare no conflict of interest.

## Notes

[1] Post-Fordisms are a variety of socioeconomic configurations in different countries of advanced production and consumption, mainly in the developed capitalist states, and came about as a response to the crisis of Fordism after the 1980s. The expanding economies of scope and different configurations in production and consumption systems and the restructuring of the welfare state in different countries were the hallmarks of this theoretical growth-crisis platform, as specialization and client specificity played a leading role in product strategy [130].

[2] Many enterprises have adopted hybrid working models, combining face-to-face and virtual work. However, remote workers have often reported a lack of satisfaction and alienation. Some have claimed that their new way of working has improved their everyday life and increased their leisure time by dramatically reducing commuting and other pertinent costs [131]. Conversely, the opponents of remote work have underlined that working outside the traditional office environment is likely to cause increased working hours and unsociable schedules, which could increase employees' psychological risk, stress, and uncertainty [132]. Also, employees from home can be obliged to pay a portion of their working expenses—e.g., broadband availability and equipment costs [133].

[3] Within traditional workplaces, the need for better health and safety standards—such as minimizing overcrowding—seems imperative for white-collar workers in the post-COVID-19 era [134].

[4] The restrictions on work mobility that occurred amid the COVID-19 crisis appear to be transferred in today's labor environment, giving rise to a novel meaning of the "workplace", in which outsourcing—and gig workers—could be a competitive advantage for the proactive employers and managers [135]. However, the degree of "telemigration" is affected by the "teleworkability" of each profession, causing a new digital divide [136].

[5] Biological analogies in economics are a widely cited concept among evolutionary and Schumpeterian economists, who suggest that biology appears more fit than mechanics to examine phenomena of reality [55,137,138].

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
