# Peer review of "The Post-COVID-19 Era, Fourth Industrial Revolution, and New Globalization: Restructured Labor Relations and Organizational Adaptation"

_societies, doi:10.3390/soc12060187_

Round 1
Reviewer 1 Report
This is a very interesting, rich, and well-referenced paper. From a methodological point of view, it follows the ‘integrative review’ perspective. However, my impression is that the paper tries to integrate too much things within the margins of an article. It is as if the authors resume here most of their previous works. This could be done in a book, but not in a Journal article. By consequent, I would propose to authors to drop one component of their integrative review and develop further the rest.
For example, they could drop the question of the organizational adaptation (micro, or strategic management level), and focus on the need of institutional innovations and how this need is translated to the specific field of labour relations. The part on the restructuring of labour relations contains interesting ideas, but deserves to be further elaborated. For example, what guarantees should labour unions have to be engaged in a bipartite/tripartite concertation process? Moreover, I think that the question of social dialogue/concertation limits de facto the whole discussion to the ‘Western countries’. Social dialogue seems unthinkable in the autocratic or unstable political regimes of the developing countries. Similar remarks apply to the equally interesting topic of institutional innovations. Figure 2 (Shaping an institutionally adaptive socioeconomic system through new forms of social Dialogue) requires more explanations and further development. Besides, the link of institutional innovations with the fourth industrial revolution and new globalisation should be elaborated further.
Minor remarks:
It is preferable to avoid long sentences and the inflationary use of ‘we contend-argue’.
Other minor remarks:
- L108: no extreme growth is expected soon. [‘robust’ or ‘steady’ instead of ‘extreme’?]
- L122: However, the 4IR only recently started to be used in relevant scholarly debates and policy forums. [‘only recently’ in the middle of the sentence]
- L154-158. Errors in the numbers of references?
- L203-205: Although Modiba and Kekwaletswe [73] 203 observe a trend toward digitization (43 percent of adults in Sub-Saharan Africa already have a bank account, up from 34 percent in 2014), [‘in comparison with’ instead of ‘up?]
- All these adverse developments are evident —primarily in Western countries but also in emerging market economies with a more significant effect— of notable trends in social and labor market inequalities. [evident of?]
- L274: especially at the micro-level of undertakings [undertakings?]
- L291-2: However, as Figure 2 illustrates, we contend that these policy approaches incorporate only to a minor extent the fact that they promote forms of institutional innovation. [ ‘the promotion’ instead of ‘the fact that they promote’?]
- L338: ‘Additionally’, or ‘Alternatively’?
Author Response
We do appreciate his/her useful comments and hereby submit our reply to Reviwer 1

Reviewer 2 Report
Referee report “Post-COVID-19 era, Fourth Industrial Revolution and the new globalization Restructured labor relations and organizational adaptation”
Summary
The paper explains the current global socioeconomic organizational changes in the context of the covid-19 pandemic, the fourth industrial revolution, and globalization. Different ongoing changes are observed on the macro, meso, and micro levels. The paper selects a range of academic publications, discusses them, and builds up its conclusions and recommendations. The author(s) conclude by stating their opinion(s) that innovation is the only way to make a socioeconomic comeback after the current recession.
In my opinion, the paper may be published under the condition that it follows the below comments.
1. Add in the title “A review and analysis of the literature”
The author(s) did not include any model to bring new findings. Hence, it is important to state in the title and the introduction that the paper reviews the literature compares its findings, and sorts on different levels (macro, meso, micro, 4IR, and Covid-19).
2. Table 1 mistake
Under 1945 to 1973, labeling that period as “US hegemony and dominance bipolarism” is a contradiction because hegemony reflects the inexistence of a clear bipolar system. Change it to Dominance bipolarism.
3. Rephrase a biased statement
“The COVID-19 crisis seems to have reignited earlier geopolitical tensions, rightly magnifying the Ukrainians’ struggle against the brutal aggression shown by Russia, whose invasion in 2022 was a culmination of the pressures that emerged in the post-COVID-19 era.”
International relations scholars and political economists would differ about such matters. What can be used as a fact is that it is an invasion made by Russia of Ukraine. However, the keywords “brutal aggression” renders the sentence biased and open for debate.
4. Important missing citation
“Also, rising health inequalities are another facet of the challenges we must face in the post-COVID-19 labor market.”. Such a sentence needs to be backed by academic findings.
Author Response
We do appreciate his/her useful comments and hereby submit our reply to Reviwer 2
